# The Effects of Hydration Parameters and Co-Surfactants on Methylene Blue-Loaded Niosomes Prepared by the Thin Film Hydration Method

**DOI:** 10.3390/ph12020046

**Published:** 2019-03-29

**Authors:** Li Key Yeo, Cheng Shu Chaw, Amal Ali Elkordy

**Affiliations:** University of Sunderland, School of Pharmacy and Pharmaceutical Sciences, Sunderland SR1 3SD, UK; bg15th@research.sunderland.ac.uk (L.K.Y.); cheng.chaw@sunderland.ac.uk (C.S.C.)

**Keywords:** niosomes, hydration time, Cremophor^®^ ELP, Span^®^ 60, thin film hydration method

## Abstract

This work aimed to investigate and optimise the effects of co-surfactants, hydration volume, and time on the entrapment of methylene blue (MB) within niosomes and the vesicle sizes of MB-loaded niosomes upon different storage temperatures. Niosomes were prepared by the thin film hydration method followed by gel permeation chromatography to obtain purified niosome suspensions. The probe sonication method was used to reduce the niosome vesicle size and distribution. Highest entrapment efficiencies (%EE) were determined for niosomal formulations containing Span^®^ 60, cholesterol, and Cremophor^®^ ELP (E2 and E3), which were prepared with a hydration volume of 5 mL. The hydration time was 15 min for E2 and 60 min for E3 (%EE = 40.1 ± 7.9% and 32.9 ± 10.1% for E3 and E2, respectively). The final lipid contents in the formulations were shown to have an impact on %EE.

## 1. Introduction

Non-ionic surfactant vesicles (niosomes) are drug carriers that have been extensively studied for delivering poorly water-soluble drugs to enhance dissolution and bioavailability [1]. To prepare a hydro-dynamically stable niosome dispersion, vesicle size, lipid content, and compositions are crucial for vesicle formation and capability retention [2,3]. 

Usually, cholesterol is used with the main non-ionic surfactant in a 1:1 molar ratio for most niosomal formulations [4]. Hence, for this current study, Span^®^ 60 and cholesterol were prepared in a 1:1 molar ratio to enable the study’s aim for investigating the effect of process parameters (hydration time and volume of niosomes) on the niosomes’ characteristics. Methylene blue (MB) was chosen as a model hydrophilic drug to be encapsulated into the prepared thin film hydrated niosomes.

MB is an antidote agent indicated for drug- or chemical-induced methaemoglobinaemia. MB-loaded niosomal formulations stabilised by either dicetyl phosphate or Solutol^®^ HS15 were previously produced by Chaw and Kim [5]. However, this study applied different stabilisers or co-surfactants (Cremophor^®^ ELP or Lauroglycol^®^ 90).

## 2. Materials and Methods

The niosomes were prepared by the thin film hydration method. Briefly, a lipid mixture of Span^®^ 60, cholesterol, and Cremophor^®^ ELP or Lauroglycol^®^ 90 was dissolved in chloroform, and the solvent was evaporated under reduced pressure (470 mbar) at 60 °C using a rotary evaporator (Buchi, Switzerland). Hydration of the film was performed by adding a Trizma buffer (pH 7.4) containing methylene blue (Sigma Aldrich, UK) to the round-bottomed flask and the flask was placed in the water bath shaker set at 100 rpm and 60 °C (refer to Table 1). Purification of niosome suspensions was carried out using a gel permeation chromatography column made up of Sephadex G50 (Sigma Aldrich, UK). The eluted niosomes were collected and disrupted using isopropanol to determine the concentration of MB using a UV–Vis visible spectrophotometer (Camspec, UK) at a wavelength of 665 nm as well as using the constructed calibration curve of MB. Vesicle sizes and size distributions of the prepared niosome dispersions were measured by a nanosize instrument (Malvern, UK) before and after probe sonication (Cole Parmer, UK) applied with an amplitude set at 40% for two cycles of two minutes each with a minute rest in between. During the sonication process, samples were placed in an ice bath to prevent overheating. This was followed by centrifugation (MSE, UK) at 5000 rpm for 5 min to remove any impurities. The vesicle sizes of the niosomes before probe sonication at two different storage temperatures (fridge, 4 °C; and room temperature, 23 °C) were determined weekly for a period of four weeks. 

Morphologies of MB-loaded niosomes were characterised by transmission electron microscopy (Hitachi, UK). A negative staining procedure was performed using a 2% aqueous phosphotungstic acid (PTA) reagent. Figure 1, which shows TEM photomicrographs, indicated that niosomes had vesicular/spherical shapes, with methylene blue in the hydrophilic cores of the prepared niosomes. 

## 3. Results and Discussion

For the prepared niosomes, hydration for a relatively short hydration time (15 min, E2) produced vesicles (before the sonication step) with larger sizes and slightly less drug entrapment efficiency compared to niosomes hydrated for a longer period of hydration (60 min, E3). This suggests that 60 min is an optimum time for the full hydration of polar heads of the used Span^®^ 60 and for the formation of vesicular spherical shapes (Table 1; Figure 1). Longer hydration time contributed to reduced vesicle size in niosomes despite different hydration volumes. However, with a constant total initial lipid mass (200 mg) used, the increasing final lipid concentration with decreasing hydration volumes resulted in a noticeable increase in %EE from E1 (22.1%) to E3 (40.1%), followed by a decrease in %EE from E3 (40.1%) to E4 (27.9%) (Table 1 and Table 2). These results concluded that a higher final lipid content (80 mg/mL) might contribute to a saturated environment where the successful formation of vesicles is prevented. Additionally, this result corresponded to the vesicle sizes of niosomes (Table 1). 

In comparison with using non-ionic water-soluble surfactant (Cremophor^®^ ELP, HLB (Hydrophilic Lipophilic Balance) value of 12–14), L1 with incorporation of non-ionic water-insoluble surfactant (Lauroglycol^®^ 90, HLB value of 3) demonstrated an increase in vesicle size but a decrease in %EE before sonication (Table 1 and Table 2). The results might be explained by the low loading and leaking of MB from the large vesicle size due to the weak membrane rigidity, as it is expected that the lipophilic Lauroglycol^®^ 90 could be competing with cholesterol in the bilayer membranes of the niosomes. Furthermore, the overall HLB for L1 is smaller compared with other formulations, and hence it has no compatibility with water-soluble MB. Therefore, further research may be carried out using Lauroglycol^®^ 90 as a co-surfactant. 

All purified formulations revealed a high polydispersity index of above 0.4, indicating non-homogeneity in size distributions before probe sonication. After probe sonication, vesicles demonstrated marked reductions in size and size distribution (Table 1), showing the effectiveness of probe sonication for producing niosomes of higher homogeneity.

Reduced vesicle size was attributed to longer hydration time. Final lipid contents in the vesicular system at 40 mg/mL demonstrated the highest %EE (Table 2). By including a probe sonication process, the sizes of non-homogenous niosome dispersions were effectively reduced, and sonication led to the formation of homogeneous niosomes. 

The stability of niosomes at different storage temperatures was assessed by changes in vesicle size (in pre-sonicated states) over time. Under fridge conditions (4 ± 2 °C), L1 niosomes showed a slight increase in size and E2 niosomes demonstrated a large variation in size over storage (Figure 2). In contrast, large fluctuations in size can be seen upon room temperature storage (23 ± 2 °C) in both E2 and L1 niosome formulations. Generally, prepared formulations which demonstrated smaller vesicle sizes revealed slight variations regardless of the different storage temperatures, with E1 showing the least variation in size.

Based on Figure 2, E3 was more stable compared to E2. E1 confirmed the results of E3 regarding good stability from using a hydration time of 60 min. E4 showed poor stability, having a high lipid content (Figure 2). L1 confirmed the E2 results of large vesicle sizes remaining the same after storage. 

## 4. Conclusions

The results of this study showed that the E3 formula with Cremophor^®^ ELP co-surfactant, 60 min hydration time, and 5 mL hydration volume produced stable niosomes with optimal lipid content for optimal MB encapsulation efficiency. The vesicle size influenced the formulation stability. 

## Figures and Tables

**Figure 1 pharmaceuticals-12-00046-f001:**
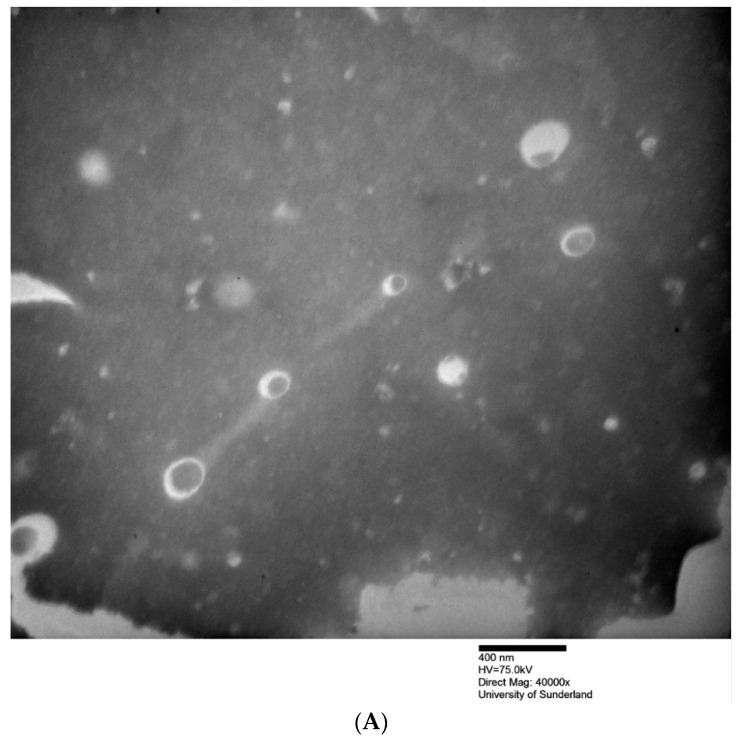
Transmission electron microscopy (TEM) images of formulations E1 (top, **A**) and E2 (below, **B**) before sonication.

**Figure 2 pharmaceuticals-12-00046-f002:**
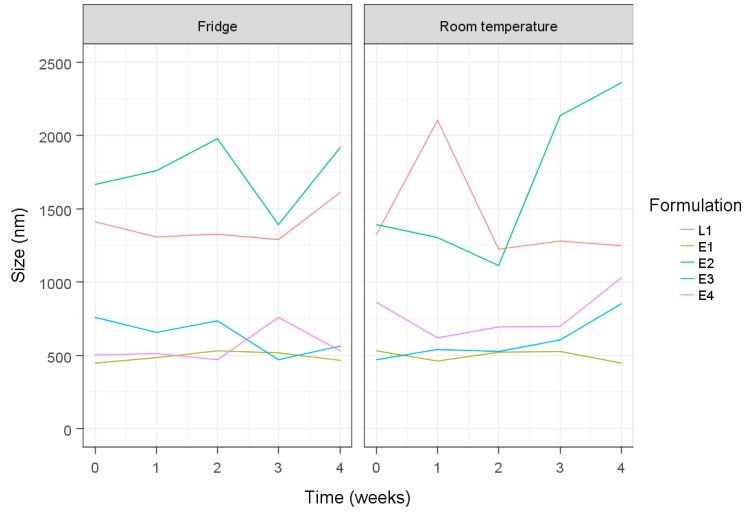
Niosome size measurements of all formulations (before probe sonication) kept under fridge (4 ± 2 °C) and room temperature (23 ± 2 °C) conditions over a four-week period.

**Table 1 pharmaceuticals-12-00046-t001:** Hydration volume (mL), hydration time (min), niosome size (nm) pre-sonication and post-sonication, and polydispersity index (PDI) pre-sonication and post-sonication of methylene blue-loaded niosomal formulations.

Formulation ^a^	Hydration Volume (mL)	Hydration Time (min)	Niosome Size (nm)		PDI	
			Before Sonication	After Sonication	Before Sonication	After Sonication
E1 S60: Cho: ELP ^b^	20	60	804.0 ± 294.8	272.9 ± 105.1	0.56 ± 0.28	0.39 ± 0.24
E2 S60: Cho: ELP ^b^	5	15	1514.3 ± 203.5	292.4 ± 12.3	0.40 ± 0.17	0.38 ± 0.23
E3 S60: Cho: ELP ^b^	5	60	615.7 ± 126.8	362.2 ± 138.3	0.71 ± 0.25	0.45 ± 0.17
E4 S60: Cho: ELP ^b^	2.5	60	619.7 ± 191.2	311.7 ± 27.4	0.72 ± 0.16	0.36 c 0.16
L1 S60: Cho: L90 ^b^	5	60	1463.4 ± 62.4	256.9 ± 51.9	0.46 ± 0.06	0.34 ± 0.13

For all experiments, results were the average of triplicates from three prepared lots (*n* = 3) ± standard deviation. ^a^ All formulations E1, E2, E3, E4, and L1 were prepared at a molar ratio of 45:45:10. ^b^ Span^®^ 60 (S60); cholesterol (Cho); Cremophor^®^ ELP (ELP); Lauroglycol^®^ 90 (L90).

**Table 2 pharmaceuticals-12-00046-t002:** Final lipid content (mg/mL) and entrapment efficiency (%EE) of methylene blue-loaded niosomal formulations.

Formulation ^a^	Final Lipid Content (mg/mL)	%EE	
		Before Sonication	After Sonication
E1 S60: Cho: ELP ^b^	10	22.1 ± 12.7	14.3 ± 1.3
E2 S60: Cho: ELP ^b^	40	32.9 ± 10.1	16.3 ± 3.5
E3 S60: Cho: ELP ^b^	40	40.1 ± 7.9	11.6 ± 4.5
E4 S60: Cho: ELP ^b^	80	27.9 ± 15.8	18.0 ± 14.5
L1 S60: Cho: L90 ^b^	40	10.5 ± 2.4	27.1 ± 1.4

For all experiments, *n* = 3. ^a^ All formulations E1, E2, E3, E4, and L1 were prepared at a molar ratio of 45:45:10. ^b^ Span^®^ 60 (S60); cholesterol (Cho); Cremophor^®^ ELP (ELP); Lauroglycol^®^ 90 (L90).

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
