# Peer review of "The Effects of Hydration Parameters and Co-Surfactants on Methylene Blue-Loaded Niosomes Prepared by the Thin Film Hydration Method"

_pharmaceuticals, 2019, doi:10.3390/ph12020046_

Round 1
Reviewer 1 Report
The brief report describes, for niosomes loaded with methylene blue, the effect of three hydration volumes, two hydration times and two surfactants on size, PDI, entrapment efficiency and size over storage on a four weeks period.
The conclusions are taken from a too small group of data: a bell-shaped trend for “hydration volume” versus “%EE” should be drawn from more than three samples (at least 5), the trend for hydration time should comprise three or four samples. The entrapment efficiency after sonication should be determined as well.
Some minor revision include:
Abstract
Page 1 rows 17 to 19: the sentence is not clear
Introduction
This is not the first paper regarding incorporation of methylene blue in niosomes by thin film hydration references doi: 10.1016/j.colsurfb.2005.09.003 and doi: 10.1111/ced.12459 should be cited
There is more recent literature on the topic that should be cited e.g: doi: 10.1016/j.cis.2013.11.018 and doi: 10.1016/B978-0-323-46143-6.00006-3
Materials and methods
Page 1 row 33 “evaporated under pressure”: what pressure?
page 1 row 40: the technique/instrument should be specified
page 2 row 46: it should be specified if the TEM images were acquired before or after sonication
page 2 rows 48-49: the sentence should be reformulated
page 2 rows 67 to 72: the hypothesis should be written in a simpler form
Author Response
Many thanks for review of the manuscript:
Response to the comments:
Methods and results sections were improved.
The brief report describes, for niosomes loaded with methylene blue, the effect of three hydration volumes, two hydration times and two surfactants on size, PDI, entrapment efficiency and size over storage on a four weeks period.
Response:
Yes
The conclusions are taken from a too small group of data: a bell-shaped trend for “hydration volume” versus “%EE” should be drawn from more than three samples (at least 5), the trend for hydration time should comprise three or four samples. The entrapment efficiency after sonication should be determined as well.
Response:
All the results included in this brief report are the average of three runs from three different preparations/lots.
Also, % EE after sonication was added, please refer to Table 2.
Some minor revision include:
Abstract
Page 1 rows 17 to 19: the sentence is not clear
Response:
The sentence was rewritten.
Introduction
This is not the first paper regarding incorporation of methylene blue in niosomes by thin film hydration references doi: 10.1016/j.colsurfb.2005.09.003 and doi: 10.1111/ced.12459 should be cited
Response:
References were added.
There is more recent literature on the topic that should be cited e.g: doi: 10.1016/j.cis.2013.11.018 and doi: 10.1016/B978-0-323-46143-6.00006-3
Response:
References were added.
Materials and methods
Page 1 row 33 “evaporated under pressure”: what pressure?
Response:
470 mbar was added.
page 1 row 40: the technique/instrument should be specified
Response:
Statements were added, lines 41 and 42.
page 2 row 46: it should be specified if the TEM images were acquired before or after sonication
Response:
Before sonication was added to Figure 1.
page 2 rows 48-49: the sentence should be reformulated
Response:
The sentence was rewritten.
page 2 rows 67 to 72: the hypothesis should be written in a simpler form
Response:
The sentences were rewritten.
Reviewer 2 Report
nice short work
Author Response
Many thanks for the supporting comments
Reviewer 3 Report
The authors report on preparation of niosomes and their use as a transporter of methylene blue. The size and morphology of niosomes were determined by DLS and TEM. Their stability was studied at room as well as low temperature.
It is not clear what the concentration of niosomes in solutions was. Is there any reason why molar ratios 45:45:10 were used? Was entrapment capacity checked after storage time? TEM figures are too large. They can be shown next to each other. It seems that conclusion of the work is missing. There is only one reference and year is not mentioned.
Author Response
Many thanks for review of the manuscript:
Response to the comments:
Introduction, Methods and results sections were improved.
- The authors report on preparation of niosomes and their use as a transporter of methylene blue. The size and morphology of niosomes were determined by DLS and TEM. Their stability was studied at room as well as low temperature.
Response:
Yes, Thanks
- It is not clear what the concentration of niosomes in solutions was. Is there any reason why molar ratios 45:45:10 were used? Was entrapment capacity checked after storage time? TEM figures are too large. They can be shown next to each other. It seems that conclusion of the work is missing. There is only one reference and year is not mentioned.
- It is not clear what the concentration of niosomes in solutions was
Response:
The total lipid content, 200mg, was added to RESULTS AND DISCUSSSION section
- Is there any reason why molar ratios 45:45:10 were used?
Response:
Usually cholesterol used with the main non ionic surfactant in a 1:1 molar ratio for most of niosomal formulations, hence it was used in this molar ratio in this study to enable the study’s aims for investigating the effect of process parameters, hydration time and volume, on niosomes characteristics. This was added into introduction section.
- Was entrapment capacity checked after storage time?
Response:
entrapment capacity has not been checked after storage time, effect of the storage time was studied on vesicle size and size distribution: results are included in this Brief Note.
- TEM figures are too large. They can be shown next to each other
Response:
TEM figure was reduced in size, however, for better visualisation of magnification and the scale, Figure was left as A, top, and B, below.
- It seems that conclusion of the work is missing.
Response:
Conclusion was added.
- There is only one reference and year is not mentioned.
Response:
More references and the year were added.
Reviewer 4 Report
Some issues should be addressed before acceptation for publications:
1. Lien 42, “This followed by” change to “this is followed by”.
2. Line 48: to delete the “images” after “Figure 1”.
3. Line 82: “with” to “and”
4. Line 89-91: The author claimed that generally formulation has smaller vesicle sizes revealing a slight variation. How could the author explain that E4 has smaller vesicle size comparing to E3, but it seems that E4 has larger variation from Figure 2?
5. More discussion about mechanisms resulting in the results are needed.
6. A conclusion section is needed.
Author Response
Many thanks for review of the manuscript:
Response to the comments:
Introduction, Methods and results sections were improved.
Some issues should be addressed before acceptation for publications:
1. Lien 42, “This followed by” change to “this is followed by”.
Response:
Done, thanks
2. Line 48: to delete the “images” after “Figure 1”.
Response:
Done
3. Line 82: “with” to “and”
Response:
Done
4. Line 89-91: The author claimed that generally formulation has smaller vesicle sizes revealing a slight variation. How could the author explain that E4 has smaller vesicle size comparing to E3, but it seems that E4 has larger variation from Figure 2?
Response:
Results after sonication from three prepared lots were added and explanation was performed and written based on both vesicle sizes before and after sonication. Accordingly, one more table was added, Table 2.
5. More discussion about mechanisms resulting in the results are needed.
Response:
More discussion added, one more table added, Table 2, as more data were required by reviewers as a result some experiments were repeated as well and Encapsulation data with Standard deviations were included in Table 2.
6. A conclusion section is needed.
Response:
Conclusion was added.
Round 2
Reviewer 1 Report
The brief report describes adequately, for niosomes loaded with methylene blue, the effect of various parameters on size, PDI, entrapment efficiency and size
Reviewer 3 Report
Thank you for the changes.
Reviewer 4 Report
It can be accepted for publication now.